# Testing strategic pluralism: The roles of attractiveness and competitive abilities to understand conditionality in men's short-term reproductive strategies

Oriana Figueroa[1,2], Jose Antonio Muñoz-Reyes[2], Carlos Rodriguez-Sickert[3], Nohelia Valenzuela[2,4], Paula Pavez[2], Oriana Ramírez-Herrera[2], Miguel Pita[5], David Diaz[6], Ana Belén Fernández-Martínez[5], Pablo Polo[2]*

1 Doctorado en Ciencias de la Complejidad Social, Centro de Investigación en Complejidad Social, Facultad de Gobierno, Universidad del Desarrollo, Concepción, Chile, 2 Laboratorio de Comportamiento Animal y Humano, Centro de Estudios Avanzados, Universidad de Playa Ancha, Valparaíso, Chile, 3 Centro de Investigación en Complejidad Social, Facultad de Gobierno, Universidad del Desarrollo, Concepción, Chile, 4 Departamento de Ciencias Biológicas, Facultad de Ciencias Biológicas, Universidad Andrés Bello, Viña del Mar, Chile, 5 Departamento de Biología, Universidad Autónoma de Madrid, Madrid, España, 6 Facultad de Ciencias Económicas, Universidad de Chile, Santiago, Chile

* pablo.polo@upla.cl

## Abstract

The decision to allocate time and energy to find multiple sexual partners or raise children is a fundamental reproductive trade-off. The Strategic Pluralism Hypothesis argues that human reproductive strategies are facultatively calibrated towards either investing in mating or parenting (or a mixture), according to the expression of features dependent on the individual's condition. This study seeks to test predictions derived from this hypothesis in a sample of 242 young men ($M \pm SD = 22.12 \pm 3.08$) from Chile's 5th Region (33° south latitude). Specifically, two predictions were considered that raise questions about the relationship between traits related to physical and psychological attractiveness (fluctuating facial asymmetry and self-perception of attractiveness) and competitive skills (baseline testosterone and self-perception of fighting ability) with short-term reproductive strategies. Our results indicate that psychological features related to the self-perception of physical attractiveness are related to short-term reproductive strategies. However, no evidence was found that fluctuating facial asymmetry, basal levels of testosterone and self-perception of fighting ability were related to short-term reproductive strategies. These results support the existing evidence of the importance of physical attractiveness in calibrating men's reproductive strategies but cast doubts about the role of fluctuating facial asymmetry. They also suggest that traits related to physical attractiveness, in comparison to competitive capabilities, play a more important role in calibrating men's short-term reproductive strategies.

**Data Availability Statement:** All relevant data are within the manuscript and its Supporting Information files.

**Funding:** PP was founded by a FONDECYT Iniciación number 11181293 from the Chilean Government. OF was founded by a scholarship for doctoral studies from the Centro de Investigación en Complejidad Social of the Universidad del Desarrollo (number 20170806005S002). The funders had no role in study design, data collection and analysis, decision to publish, or preparation of the manuscript.

**Competing interests:** The authors have declared that no competing interests exist.

## Introduction

Reproductive strategies can be defined as an integrated set of adaptations that constitute solutions to different reproductive compromises or trade-offs that the individual faces [1]. The Strategic Pluralism Hypothesis seeks to explain inter- and intra-individual variation in human reproductive strategies based on the expression of traits dependent on the condition of the individual in interaction with the environment. This hypothesis emphasizes the costs and benefits for men and women concerning the resources invested in seeking partners versus providing parental care [1, 2]. In mammals, the trade-off for males between the search for a partner and parental care is particularly relevant given that males invest less than females in obligatory parental care while having a higher potential reproductive rate [3]. This means that for males, including men, maximizing reproductive success is mainly constrained by the degree of access to multiple reproductive partners [3, 4]. However, biparental care in humans may represent an important factor that affects offspring survival and, then, constrains men's reproductive success as well [5]. According to the Strategic Pluralism Hypothesis, individuals would display a mixture of short and long-term reproductive strategies, reflecting different degrees of investment in mating versus parenting effort according to individual phenotypic features and ecological and social conditions [1]. In this sense, the ability to attract partners and to compete with individuals of the same sex are two factors that affect the degree of investment in short-term reproductive strategies, given that they reduce the costs-benefits balance of investing in mating [6]. Consequently, the integrated study of these two factors is key to understanding their influence on short-term reproductive strategies.

Physical attractiveness is directly related to the capacity of being chosen as a mate [6–8]. Research in this field has identified a series of bodily features associated with attractiveness, facial symmetry being one of these [9, 10]. Fluctuating facial asymmetry has been proposed as an indicator of individual quality that reflects the capacity of an individual to maintain a symmetric pattern of stable development [9, 11]. However, there is mixed evidence supporting the notion of fluctuating asymmetry as a reliable indicator of the degree of developmental stability and individual quality [12–14]. In turn, fluctuating facial asymmetry is assumed to underlie individual differences in facial symmetry and, consequently, in attractiveness, but this assumption is rarely tested. Despite this, some studies suggest that men with relatively lower levels of fluctuating facial asymmetry are more attractive to women, especially for short-term relationships, are more economically successful, less faithful, and less inclined to invest in their progeny [15–18]. These results are consistent with evidence that men with lower levels of fluctuating facial asymmetry have more sexual partners and tend to be more direct in approaching the opposite sex in courtship, a characteristic that is related to short-term or unrestricted reproductive strategies [1, 8, 14, 19, 20]. Nevertheless, other investigations [10, 21] have not been able to replicate the association between the number of sexual partners and fluctuating facial asymmetry in men. These contradictory findings indicate the need to generate new studies in the field including psychological variables that may influence the relationship between fluctuating facial asymmetry and reproductive success to further test the relevance of fluctuating facial asymmetry as a signal of quality. Particularly relevant for our study is the relationship that has been observed between self-perception of physical attractiveness and the prevalence of short-term reproductive strategies since there is an association between self-perceived characteristics and received social signals, which together affect behavior. That is, the assessment of one's attractiveness is associated with short-term strategy because it reflects the preference of women for certain traits [6, 22, 23].

At the level of competitive abilities, intrasexual competition is another component that influences access to partners of the opposite sex [6, 24]. Testosterone is related to the

development of the traits and behaviors related to intrasexual competition, as the display of direct physical aggression [25]. Testosterone is an androgenizing hormone with two main types of effects: organizational and activational. At the organizational level, it has an androgenizing effect during the prenatal stage and at puberty. At the activational level, baseline testosterone levels, as well as changes in circulating testosterone levels, have been associated with behavioral changes related to intrasexual competition and reproductive effort [26–28]. Focusing on baseline testosterone, there is evidence that baseline testosterone levels are positively related to dominance, competitiveness [28, 29], and especially aggression [30], although this effect depends on the specific context [27], and some studies reported a lack of evidence for the mentioned relationships [31]. Lower baseline testosterone levels have been associated with men's relational status and paternity; that is, baseline testosterone levels are lower in men involved in long-term relationships, and especially in those that are fathers [32, 33, for a review see 34]. However, these levels can be expected to remain high in men who, although involved in a relationship, are interested in having extramarital relationships [35–37]. Puts et al. [38] studied the relationship between testosterone levels and three dimensions of sociosexuality: sociosexual desire, sociosexual behavior, and sociosexual attitudes. Their results indicate that testosterone levels are positively related to unrestricted sociosexual psychology (desires and attitudes), which results in a larger number of reproductive partners (behavior). However, they found that the number of reproductive partners has a negative effect on testosterone levels. They interpret these findings as a negative feedback mechanism that prevents maintaining high testosterone levels once sociosexual desires have been satisfied [38]. This suggests that testosterone plays an important role in the willingness of individuals to compete for reproductive partners, which in turn implies searching for short-term strategies. This relationship is complex because there are negative regulatory mechanisms. At the psychological level, testosterone increases competitive behavior and reduces cooperation in determined contexts [29] suggesting that testosterone levels influence intrasexual competition and reproductive success [28, 36, 39]. In addition, the self-perception of having traits related to resource holding potential, like being a good fighter, may be positively related to signs of dominance in competitive contexts [40] suggesting that this psychological feature plays a role when competing for mating.

Competitive abilities and physical attractiveness do not act independently. Lukaszewski et al. [23] examined the effect of body strength as an indicator of both fighting ability and attractiveness, and the self-perception of physical attractiveness, as well as the assessment by third parties of physical attractiveness and sociosexual attitudes and behaviors. Their results show that self-perception of physical attractiveness mediates the positive effect between physical strength, unrestricted socio-sexual attitudes, and the number of sexual partners. This indicates the need to explore how indicators of physical attractiveness and fighting ability explain unrestricted human reproductive strategies, using a larger number of morphological, physiological and psychological indicators.

Considering all of the above, we can establish that attractiveness and competitive abilities are important elements that according to the Strategic Pluralistic Hypothesis are expected to play a major role calibrating unrestricted reproductive strategies, especially when a combined effect of the two is displayed. However, there have been few studies that consider biological (morphological and physiological) and psychological variables in an integrated manner in order to understand how reproductive trade-offs are dealt with. The objective of this study is to investigate how the features of physical attractiveness and competitive abilities influence short-term male reproductive strategies and how psychological features may act as moderators of these effects. To do this, fluctuating facial asymmetry and the self-perception of physical attractiveness are considered as anthropometric and psychological features of physical attractiveness. Likewise, the levels of circulating baseline testosterone and self-perception of fighting

ability are considered physiological and psychological features associated with competitive capacity. Considering the postulates of the Strategic Pluralism Hypothesis, we should expect a positive association between features signaling physical attractiveness and competitive capacity with short-term reproductive strategies because both sets of traits decrease the costs associated with seeking and competing for mates and satisfy the short-term mating preferences of women. Our particular predictions are as follows: (1) fluctuating asymmetry should be negatively associated with short-term reproductive strategies, especially in individuals with high levels of self-perceived physical attractiveness, whereas basal levels of testosterone should be positively associated with short-term strategies, especially in those individuals with high levels of self-perceived fighting ability; (2) the individual's competitive abilities are expected to moderate the effect of attractiveness on short-term reproductive strategies. In this way, attractiveness has a positive effect on short-term reproductive strategies that is greater for individuals who display higher competitive abilities.

## Materials and methods

### Ethics statement

The research was approved by the ethics committee of Universidad de Playa Ancha. Participants signed a written informed consent before they participated in the study.

### Participants

The initial sample was composed of 246 young men. However, three individuals were rejected because they failed to complete some of the questionnaires and one individual failed to provide a photo, so the sample was reduced to 242 men between 18 and 35 years of age ($M \pm SD$ = 22.12 ± 3.08). The participants were recruited using ads posted in universities in the 5th Region of Chile (33˚ south latitude). In terms of sexual orientation, 97.5% stated they were heterosexual and 3.5% stated they were homosexual. In terms of relational status, 53.8% stated they were in a couple at the time of participating in the investigation.

The participants received an economic reimbursement of five thousand Chilean pesos (approximately seven US dollars) as an economic reimbursement for participating, plus up to 30,000 pesos (approximately 43 USD and almost twice of the daily minimum wage) conditional on their performance in the economic games that were played as a part of a wider experimental procedure. These games were introduced and played after the measurement of basal testosterone and after the participants answered the questionnaires. Consequently, these games were not expected to affect responses and measures considered in this study.

### Psychological measurements

**Sociosexual Orientation Scale (SOI).**  We used a multidimensional version of the SOI developed by Jackson & Kirkpatrick [41] that had been applied previously with Chilean subjects [see 42]. The scale is divided into attitudinal and behavioral dimensions. There are two attitudinal factors that measure sociosexual orientation in the short-term (e.g. "I can easily imagine being comfortable with and enjoying casual sex with different women", 10 items) and long-term (e.g. "I am interested in maintaining a long-term romantic relationship with a special woman", 7 items). These factors are in the format of 7-point Likert scale responses in which 1 indicates "*strongly disagree*" and 7 indicates "*strongly agree*". The behavioral dimension consisted of 5 items of open-ended responses that included questions about the number of sexual partners in the past (3 items) (e.g. "Over your entire life, how many women have you had complete sexual relations with?), a question about sexual fantasies *(*How often do you

fantasize about having sexual relations with women other than your current partner?*)* and a question about the expected number of sexual partners in the future *(*How many women do you think you will have sexual relations within the next five years*)*. This study only considered the items referring to the attitudinal factor in short-term relationships. Polo et al. [42] obtained a Cronbach α value of .95 for the aforementioned factor, while a Cronbach α of .70 was obtained in the present study, indicating that the instrument is sufficiently reliable.

**Self-perceived fighting ability questionnaire.**   We used a version of the self-perceived fighting ability questionnaire developed by Muñoz-Reyes et al. [25], which had been applied previously with Chilean subjects [see 43]. This is a short 4-question questionnaire that assesses the self-perception of fighting skills (1. How good a fighter am I? 2. How do others perceive my abilities as a fighter? 3. How much fear can I provoke in someone who is about to fight me? 4. What are my odds of winning a fight if I have to fight someone?). The responses are on a seven-point Likert scale in which 1 indicates "*well below average*" and 7 indicates "*well above average*". Muñoz-Reyes et al. [43] obtained a Cronbach α score of .84 in the original study. In the present study, an α coefficient of .87 was obtained, indicating adequate reliability for the studied sample.

**Self-perception of attractiveness.**   This consists of a single question to assess self-perception of physical attractiveness (How attractive do you think you are?). The response is on a 7-point Likert scale in which 1 indicates "*not attractive at all*" and 7 indicates "*very attractive*".

## Anthropometric and physiological measurements

**Fluctuating facial asymmetry.**   This indicator of attractiveness is measured according to the protocol of Sanchez-Pages & Turiégano [44]. Frontal photographs were taken of the participants with a Nikon D-90 camera under constant conditions of light, head orientation, focal length (3 m), shutter speed (1/60 s) and aperture (f/5.6). Participants were asked not to wear any form of facial adornment and to maintain as neutral an expression as possible. Photos where the subject smiled or inclined his head were rejected and we selected the best photo of each participant. Fluctuating facial asymmetry was calculate based on 106 facial points or landmarks (LM), which were obtained with the program FACE ++ [see 45 for a similar procedure, 46] from the selected photos. This software identifies high-precision facial reference points, like facial contours, eyes, eyebrows, and nose. The use of this software was automated with a MatLab software algorithm connected to the interface of programming applications of FACE ++. Fluctuating facial asymmetry was determined with the software MorphoJ [47] (also see http://www.flywings.org.uk/MorphoJ_page.htm) based on the Procrustes distances between each LM the corresponding mirror image LM. These distances reflect both directional and fluctuating asymmetry but can be decomposed in these two components employing a Procrustes ANOVA analysis since Procrustes coordinates are based on the algebra of sums of squares [48, 49]. In this sense, the variance attributable to the variable "side of the face" corresponds to directional asymmetry, whereas the variance attributable to the interaction between "side of the face" and "individual" corresponds to fluctuating asymmetry. In other words, fluctuating asymmetry was calculated as the deviation of each individual's asymmetry from the overall average asymmetry in units of Procrustes distance. Accordingly, higher values represent higher levels of individual fluctuating asymmetry than lower values. In addition, we calculated the distribution of the differences between each LM and the corresponding mirror LM from each individual and each coordinate in order to characterize the nature of the variation in the asymmetry component. We found that differences between LM in the horizontal axis were normally distributed in 44 out of 48 pairs of LM, whereas differences between LM in the vertical axis were normally distributed in 45 out of 48 pairs of LM.

**Baseline testosterone.**   Baseline testosterone was measured with a 1-ml sample of saliva of the participants, who were asked not to eat or drink anything other than water for at least one hour before the sample was taken. The samples were taken at approximately noon to avoid alterations in testosterone levels as the result of circadian hormonal variation. A passive saliva collection method (Salimetrics®) was used to collect samples. After being collected, the samples were centrifuged, frozen and stored at -20˚C in cyrotubes (SalivaBio®) for 20 days. All the samples were analyzed with the Testosterone Enzyme Immunoassay Kit (Salimetrics®) in accordance with manufacturer's instructions. Due to a problem with storing the saliva samples (freezing rupture and consequent increase in temperature), 136 individuals were not included in the baseline testosterone analysis because the intra and interplate variation coefficients were very high. Thus, the sample for all the baseline testosterone analyses was 106 individuals between 18 and 35 years of age ($M \pm SD$ = 22.34 ± 3.08). The coefficients of intraplate and interplate variation were respectively equal to and less than 15%.

## Statistical analysis

To test our predictions, we fitted one general linear model in three successive steps. The two first steps tested our first prediction and the third step tested our second prediction. Fluctuating facial asymmetry, basal levels of testosterone, self-perception of physical attractiveness and self-perception of fighting ability were considered independent variables, and short-term socio-sexual orientation was considered the dependent variable. The age and relational status of the subjects were also considered as control variables. Independent and control variables were centered on their means. In a first step, we fitted only the main effects in the model. In the second step, we added the interaction terms between fluctuating asymmetry and self-perception of physical attractiveness and between basal levels of testosterone and self-perception of fighting ability. In this way, we tested our first hypothesis taking into account that nonsignificant interactions may preclude to assess main effects. Finally, we added the interaction term between fluctuating asymmetry and basal levels of testosterone and between self-perception of physical attractiveness and self-perception of fighting ability to test our second prediction. Because of the reduced sample size with the introduction of the baseline testosterone variable and because it did not have a statistically significant effect on the model, the variable was eliminated to restore the complete sample. A similar analytical strategy was used to test our predictions but without basal testosterone levels and using the full data set. Consequently, we finally fitted two models, one with the reduced data set and the other one with all the data available, in three successive steps.

The normality of residuals was verified for the two models and IBM SPSS 21.0 software was used for the general linear models. The level of significance was set at $\alpha$ = .05.

## Results

Table 1 shows the descriptive statistics for the morphological, physiological and psychological variables. First, we show the results considering the reduced data set, that is, the model that considers baseline testosterone levels. We did not find any significant main effect of our variables (see Table 2). When introducing the predicted interactions between physical and psychological variables, we found that self-perception of physical attractiveness did not moderate the predicted relationship between fluctuating facial asymmetry and unrestricted sociosexual orientation ($B$ = -236.910, $t$ = -.685, $p$ = .495). Similarly, the self-perception of fighting ability did not moderate the predicted relationship between basal testosterone levels and unrestricted sociosexual orientation ($B$ = .002, $t$ = .473, $p$ = .638). Finally, when we considered the interaction between attractiveness and competitive traits at both levels (physical and psychological),

**Table 1. Descriptive statistics for the variables employed in this study.**

|  | Reduced sample (N = 106) | | Full sample (N = 242) | |
|---|---|---|---|---|
| **Variable** | **M ± SD** | **Range (min, max)** | **M ± SD** | **Range (min, max)** |
| Age | 22.34 ± 3.08 | 18, 35 | 22.12 ± .3.08 | 18, 35 |
| FFA | .016 ± .005 | .008, .033 | .016 ± .005 | .008, .033 |
| Basal testosterone | 212.18 ± 75.91 | 94.31, 506.80 |  |  |
| SPPA | 4.77 ± .99 | 2, 7 | 4.68 ± .97 | 1, 7 |
| SPFAQ | 17.26 ±.4.78 | 5, 28 | 16.80 ± 4.88 | 4, 28 |
| Short-term SOI | 44.63 ± 13.62 | 10, 70 | 44.45 ± 13.42 | 10, 70 |

Fluctuating facial asymmetry (FFA), self-perception of physical attractiveness (SPPA), self-perception of fighting ability (SPFA), short-term sociosexual orientation inventory (Short-term SOI).

we did not find that the interaction terms were significant, that is, basal testosterone levels did not moderate the predicted relationship between fluctuating facial asymmetry and unrestricted sociosexual orientation ($B = 1.783$, $t = .435$, $p = .665$), and self-perception of fighting ability did not moderate the predicted relationship between self-perception of physical attractiveness and unrestricted sociosexual orientation ($B = .296$, $t = 1.125$, $p = .263$). Overall, our results with the reduced data set showed that no variable was related to unrestricted sociosexual orientation neither when considering main effects nor the predicted interactions.

When considering the full data set, that is, setting aside the basal levels of testosterone, we found a positive relationship between the self-perception of psychical attractiveness and unrestricted sociosexual orientation when fitting main effects ($B = 2.074$, $t = 2.215$, $p = .028$; see Table 3 and Fig 1). However, neither fluctuating facial asymmetry ($B = 208.547$, $t = 1.104$, $p = .271$) nor self-perception of fighting ability ($B = .195$, $t = 1.053$, $p = .293$) were related to unrestricted sociosexual orientation. Moreover, self-perception of psychical attractiveness did not moderate the predicted relationship between fluctuating facial asymmetry and unrestricted sociosexual orientation ($B = -203.254$, $t = -.945$, $p = .346$). And finally, self-perception of fighting ability did not moderate the predicted relationship between self-perception of psychical attractiveness and unrestricted sociosexual orientation ($B = .309$, $t = 1.742$, $p = .083$). Overall, our results with the full data showed a positive effect of self-perception of psychical attractiveness over unrestricted sociosexual orientation being the other predicted effects non-significant.

**Table 2. General linear model for short-term sociosexual orientation considering the reduced data set (N = 106).**

| Model | $R^2_{adj}$ | $p$ |  | $B$ | $t$ | $p$ | $\eta^2_p$ |
|---|---|---|---|---|---|---|---|
| Main effects | .041 | .116 | Intercept | 47.568 | 22.937 | < .001 | .842 |
|  |  |  | RS = Paired | -5.107 | -1.814 | .073 | .032 |
|  |  |  | Age | .306 | .708 | .481 | .005 |
|  |  |  | Basal testosterone | .011 | .611 | .543 | .004 |
|  |  |  | FFA | 190.731 | .691 | .491 | .005 |
|  |  |  | SPPA | 1.594 | 1.106 | .271 | .012 |
|  |  |  | SPFA | .535 | 1.759 | .271 | .012 |
| Interaction terms (1) | .028 | .213 | FFA * SPPA | -236.910 | -.685 | .495 | .005 |
|  |  |  | Basal testosterone * SPFA | .002 | .472 | .638 | .002 |
| Interaction terms (2) | .037 | .167 | FFA*Basal testosterone | 1.783 | .435 | .665 | .002 |
|  |  |  | SPPA*SPFA | .296 | 1.125 | .263 | .013 |

Relational status (RS), fluctuating facial asymmetry (FFA), self-perception of physical attractiveness (SPPA), self-perception of fighting ability (SPFA).

**Table 3. General linear model for short-term sociosexual orientation considering the full data set (N = 242).**

| Model | $R^2_{adj}$ | p | | B | t | p | $\eta^2_P$ |
|---|---|---|---|---|---|---|---|
| Main effects | .050 | .004 | Intercept | 46.532 | 37.430 | < .001 | .856 |
| | | | RS = Paired | -3.870 | -2.271 | .024 | .021 |
| | | | Age | .493 | 1.772 | .078 | .013 |
| | | | FFA | 208.547 | 1.104 | .271 | .005 |
| | | | SPPA | 2.074 | 2.215 | .028 | .020 |
| | | | SPFA | .195 | 1.053 | .293 | .005 |
| Interaction terms (1) | .049 | .006 | FFA* SPPA | -203.254 | -.945 | .346 | .004 |
| Interaction terms (2) | .058 | < .001 | SPPA*SPFA | .309 | 1.742 | .083 | .013 |

Relational status (RS), fluctuating facial asymmetry (FFA), self-perception of physical attractiveness (SPPA), self-perception of fighting ability (SPFA).

## Discussion

The Strategic Pluralism Hypothesis explains the conditionality of human reproductive strategies and the resolution of the trade-off between investment in multiple partners and investment in parental care [1]. This hypothesis considers that there are biological, psychological and anthropometric factors that calibrate reproductive behavior according to the context in which the individual faces the aforementioned trade-off. This study proposes two predictions that were mainly not sustained as only an effect of self-perceived physical attractiveness on short-term sociosexual orientation was found. Our results emphasize the role of physical attractiveness in men on the unfolding of unrestricted reproductive strategies (short-term strategies at the scale of sociosexual orientation). The main result indicates that the traits of attractiveness have an effect on unrestricted reproductive strategies whereas fighting abilities do not.

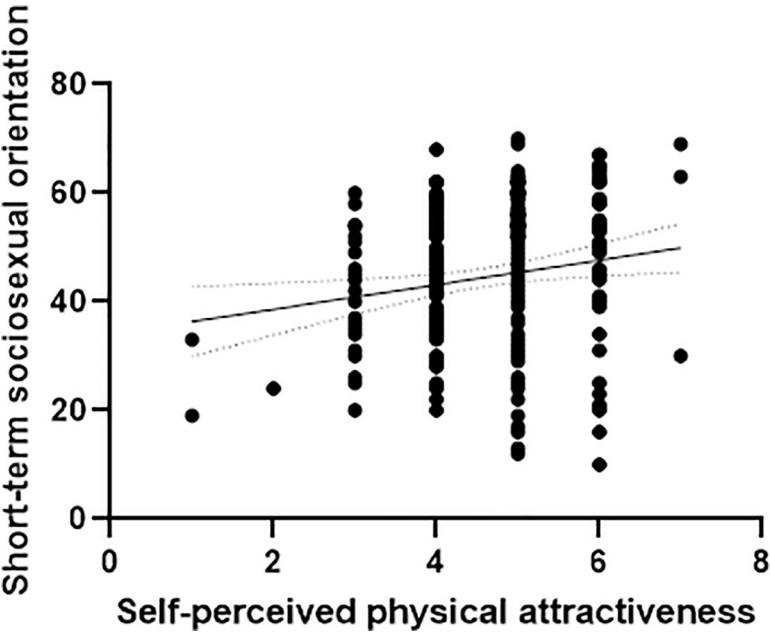

**Fig 1. Relationship between self-perceived physical attractiveness and short-term sociosexual orientation.** Dots represent observed values. Full line represents expected values across the observed range of short-term sociosexual orientation. Dotted lines represent 95% interval confidence bands of the predicted values.

The first prediction sought to determine if there is a positive relationship between traits associated with physical attractiveness and traits associated with competitive abilities with unrestricted reproductive strategies. Also, we postulated that these effects should be moderated by psychological variables related to self-perception of physical attractiveness and fighting ability, respectively. Our results suggest that only self-perceived attractiveness does affect unrestricted sociosexual orientation when evaluated with the full data set. However, we failed to show the expected effect of fluctuating facial asymmetry on unrestricted reproductive strategy either as a main effect or moderated by self-perception of physical attractiveness. Conversely to our results, some studies found evidence that fluctuating facial asymmetry is associated with the implementation of short-term reproductive strategies [20], with the number of sexual partners over one's lifetime [50, 51], and with the perception of attractiveness [14]. Several lines of evidence may explain our contrasting results. First, despite that symmetry is associated with attractiveness, this association is weak and other facial features like averageness may play a more important role in perceived attractiveness [52]. Also, facial fluctuating asymmetry is assumed to underlie variation in facial symmetry between individuals being an indirect measure of overall symmetry. This circumstance may lessen its relationships with attractiveness. And more importantly, there exists controversy about the relationship between evolutionary relevant features and levels of fluctuating facial asymmetry [21]. If it is the case that fluctuating asymmetry is not an accurate proxy of developmental instability, the rationale about the importance of fluctuating facial asymmetry as a trait related to short-term mating strategies weakens and other variables as muscularity or strength could be better predictors of an unrestricted sociosexual orientation than fluctuating facial asymmetry [42, 53]. However, this is an unresolved issue as positive evidence about the importance of fluctuating asymmetry as a proxy of health and mating success was also reported [14]. An alternative explanation of our results is that fluctuating facial asymmetry affects unrestricted sociosexual orientation but this effect is mediated rather than moderated by psychological features. In this regard, previous investigations have found that the effects of morphological features on the psychology of unrestricted male sociosexual behavior were mediated by self and third-party perceptions of physical attractiveness [23, 54]. However, our cross-sectional design precluded us to investigated mediation relationships in an accurate way [55].

Self-perception of fighting ability was not related to unrestricted sociosexual orientation. In this regard, other studies have established a relationship between fighting and mate value [43], which is defined as "the complete set of characteristics that an individual has in a given moment and in a particular context that affects his capacity to successfully find, attract and keep a partner" [56]. According to Muñoz-Reyes et al. [43], fighting ability is associated with the mate value of a partner, which implies a positive relationship between this variable and men's assessments of their chances of finding partners, and therefore of employing intrasexual competition strategies, which implies a high degree of self-confidence in the search for partners. It has been established that the self-perception of fighting ability is also associated with aggressive behavior [25]. These findings indicate that it is plausible to support that fighting ability is a conflict resolution mechanism in situations of intrasexual competition, which is consistent with studies that have found a positive association between traits associated with fighting abilities and reproductive success [23, 43, 57]. Despite the above evidence, our null results may indicate that self-perception of fighting ability when evaluated jointly with self-perception of attractiveness is not an important factor related to unrestricted strategies. That can be explained if we assume that mate choice or indirect competition through showing off attractive features may be more important in industrialized societies rather than the direct competition through fights.

No effect was found for baseline testosterone levels on short-term reproductive strategies. Studies have associated testosterone with the search for social status [58], self-confidence in

competitive situation [39] and the adoption of dominant roles in economic environments [29]. Consequently, testosterone can be considered a social hormone associated with status-seeking and not so much with aggression in itself. Status in turn could be related to different reproductive strategies according to the way it is acquired. The relationship between testosterone and reproductive strategies has been explored in other studies and evidence has been found that favors the role of testosterone as a promoter of short-term strategies. For example, Edelstein et al. [32] found an interaction between unrestricted sociosexuality and the relational status of men and established that men in relationships with partners, but that have interest in extramarital relationships, have similar testosterone levels as those of single men, producing a positive attitude about unrestricted strategies. Puts et al. [38] established that there is a negative relationship between the number of sexual partners and baseline testosterone levels, and a positive relationship between high levels of baseline testosterone and unrestricted sociosexual psychology (desire and attitudes). Although this investigation employed a reduced version of the sociosexual orientation questionnaire [59], a relationship was found between baseline testosterone levels and an orientation toward short-term strategies. The reduced sample in the model that assesses the effect of baseline testosterone on reproductive strategies could explain the null result with respect to this variable.

Based on the second prediction, a relationship was expected between attractiveness and competitive abilities on unrestricted strategies. We failed to find that association as the effect of self-perception of attractiveness was not moderated by self-perception of fighting abilities. This result further suggests that fighting abilities do not play a major role in unrestricted sociosexual orientation both directly or moderating the effects of physical attractiveness. In addition, it is important to consider that self-perception of fighting ability may not necessarily be related to the willingness to compete for new mates, but may be also associated with the willingness to protect a current mate and the offspring. In this regard, that feature is expected to be related to more restricted sociosexual orientation reflecting a higher inversion in parental care [6].

Among the limitations of this research is the inclusion of only one anthropometric measurement (fluctuating facial asymmetry), which, although a common measurement to study physical attractiveness, could be complemented with others that are also considered attractive features and, in some cases, more important in explaining facial attractiveness [52]. Another limitation was the loss of data due to the storage of samples and handling of the testosterone kit, despite following protocols tested in other investigations. In addition, our null results of the effect of basal testosterone on sociosexuality do not preclude a potential relationship between testosterone changes elicited on a mating context and sociosexuality. Changes in testosterone levels and additional anthropometric variables associated with unrestricted strategies should be included in future research, such as facial masculinization [35], height [e.g. 60] and body mass [e.g. 42]. Finally, individuals in our study expected to participate in a competitive game and to be paid according to their performance. Despite that these tasks were performed after the measures taken for the current study; they may introduce some noise in the study.

In conclusion, the present study contributes some evidence that supports the Strategic Pluralism Hypothesis as we found that psychological features of attractiveness are related to unrestricted reproductive strategies among men. However, our results are not conclusive about the potential role of competitive skills (measured by basal levels of testosterone and self-perception of fighting ability) and the role of fluctuating facial asymmetry in explaining unrestricted reproductive strategies. These findings encourage further research on traits that may be affecting the cost-benefits balance in the reproductive trade-off that men have between maximizing the number of sexual partners and investing in parental care, and designs that allowed to investigate mediation relationships considering the importance of the relationship among

anthropometric features on the self-perception (that is, psychological features) of subjects when the reproductive trade-off is solved.

## Supporting information

**S1 Dataset.**
(XLSX)

**S1 File. Normality and t tests.**
(PDF)

## Acknowledgments

We thank two anonymous reviewers and the editor for their valuable comments on earlier versions of the manuscript.

## Author Contributions

**Conceptualization:** Oriana Figueroa, Jose Antonio Muñoz-Reyes, Pablo Polo.

**Data curation:** Jose Antonio Muñoz-Reyes, Oriana Ramírez-Herrera, Pablo Polo.

**Formal analysis:** Nohelia Valenzuela, Pablo Polo.

**Funding acquisition:** Pablo Polo.

**Investigation:** Jose Antonio Muñoz-Reyes, Nohelia Valenzuela, Paula Pavez, Oriana Ramírez-Herrera, Miguel Pita, David Diaz, Ana Belén Fernández-Martínez.

**Methodology:** Pablo Polo.

**Writing – original draft:** Oriana Figueroa.

**Writing – review & editing:** Oriana Figueroa, Jose Antonio Muñoz-Reyes, Carlos Rodriguez-Sickert, Paula Pavez, Pablo Polo.

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
