## [Decision Letter · Decision Letter 0]

4 Mar 2020

PONE-D-19-27652

Testing strategic pluralism: The roles of attractiveness and competitive abilities to understand conditionality in men’s short-term reproductive strategies

PLOS ONE

Dear Dr. Polo,

Thank you for submitting your manuscript to PLOS ONE. After careful consideration, we feel that it has merit but does not fully meet PLOS ONE’s publication criteria as it currently stands. Therefore, we invite you to submit a revised version of the manuscript that addresses the points raised during the review process.

Dear Dr Polo,

Thank you for submitting your work to PLOS ONE, and apologies for the delay in returning the manuscript to you. I have now secured the opinion of two reviewers in the field. As you will see from their comments, they generally find much to like in the manuscript, commenting on the large sample size, the number of theoretically driven measures, and the testing of an influential theory in the area. They also highlight different concerns about the manuscript, which I agree with. The concerns of Reviewer 1 centred around the presentation of the hypotheses of the SPH, which was presented as an essentially dichotomous outcome for a given male, when in reality it is likely to be more continuous. The bearing on the results and interpretation should be considered. Reviewer 1 also raised concerns over the measurement, definition, and accuracy of the use of facial asymmetry as a proxy for attractiveness perception. This is something I also agree with, and have researched myself (e.g. Jones and Jaeger, 2019; Symmetry). I do not think that these are insurmountable issues and careful discussion and interpretation of the measures is required.

Reviewer 2 pointed out some more statistical issues with the manuscript, as well as the use of economic games which was puzzlingly mentioned once, and then is not touched upon again. I agree with Reviewer 2 in that much more detail is needed about the background of the work here. In my opinion, and after reading the manuscript, perhaps the most problematic part of the manuscript were the analyses and the modelling procedure. Reviewer 2 also pointed this out and I am in full agreement with them on this point; that there are too many models here testing too many hypotheses.

The main issue is that while you have sufficient data and variables to test a model of short-term SOI approaches, splitting variables across into different models obviously ignores the fact that in reality, these variables were measured from, and exist within, the sample of males you obtained. The data-generating process (DGP) is not being accurately tested here. In my view, you have the capacity to run build two models - one which includes all predictors and your full sample, which includes any theoretically important interactions that you wish to test, and another which is includes testosterone as a measure on the reduced sample for which those measures are available for. This way, you can be more certain in your conclusions about what predictors affect what in the presence of other variables, rather than splitting things up. Alternatively, you may wish to first account for the variation explained by age and partner status, and fit your models in a stepwise fashion. Either way, I do not think the current analysis is convincing in its conclusions as it fails to approximate the DGP. I also wondered about the transformations used here. Squaring a variable if it is somewhat non-normal will not make it more normal, but will lead to a kind of gamma-distributed variable (most values bunched up to one side of the distribution), but in any case the mathematics of ordinary least squares does not mind non-normally distributed variables for the predictors or the response; only that the residuals are normally distributed. This is mentioned implicitly but I couldn't tell if the residuals were normal or something was wrong with the predictors.

I was unsure about the use of the mediation analysis here. There are significant reservations about the use of mediation analysis in cross sectional designs - for mediation to be conclusive multiple time points are required, or at least with a cross sectional design heavily theoretical arguments and multiple measures used as mediators should be used to build confidence in the results. I think a far safer conclusion could be made by specifying an interaction between FA and SPA - for example, do men with lower levels of FA perceive themselves to be particularly attractive, and thus have a specific SOI score? To test this would require a simple interaction and centring of your variables. 

The manuscript was well written and has a high quality data set (which I would encourage the authors share via the OSF, if they can), and I look forward to receiving a resubmission that addresses these concerns.

We would appreciate receiving your revised manuscript by Apr 18 2020 11:59PM. To enhance the reproducibility of your results, we recommend that if applicable you deposit your laboratory protocols in protocols.io, where a protocol can be assigned its own identifier (DOI) such that it can be cited independently in the future. For instructions see: http://journals.plos.org/plosone/s/submission-guidelines#loc-laboratory-protocols

We look forward to receiving your revised manuscript.

Kind regards,

Alex Jones

Academic Editor

PLOS ONE

Journal Requirements:

3. Please amend your list of authors on the manuscript to ensure that each author is linked to each affiliation. Authors’ affiliations should reflect the institution where the work was done (if authors moved subsequently, you can also list the new affiliation stating “current affiliation:….” as necessary).

Reviewers' comments:

Reviewer's Responses to Questions

**Comments to the Author**

1. Is the manuscript technically sound, and do the data support the conclusions?

Reviewer #1: Partly

Reviewer #2: Partly

2. Has the statistical analysis been performed appropriately and rigorously? 

Reviewer #1: Yes

Reviewer #2: Yes

3. Have the authors made all data underlying the findings in their manuscript fully available?

Reviewer #1: Yes

Reviewer #2: Yes

4. Is the manuscript presented in an intelligible fashion and written in standard English?

Reviewer #1: Yes

Reviewer #2: No

5. Review Comments to the Author

Reviewer #1: This study gathers together evidence in support of the Strategic Pluralism Hypothesis, the idea that males can allocate time and resources to either mate selection or the raising of children. The study is well done and believable, but the paper can be improved especially in the presentation of the Strategic Pluralism Hypothesis and its predictions.

The Strategic Pluralism Hypothesis is presented first, followed by a set of the sub-hypotheses. These sub-hypotheses are actually predictions of the Strategic Pluralism Hypothesis, but is unclear how they are related. They also need to be portrayed as predictions. For example, if SPH is true, then we expect a, b, and c to be true. Finally, what are the alternatives to the SPH? Science functions best when there are alternative explanations for phenomena.

The Strategic Pluralism Hypothesis is posed as an either/or affair, when it is really an entire spectrum. The two ends of the spectrum are 100% of the time and energy spent trying to mate with every available partner versus 100% of the time spent raising children with one partner. But the time and energy allocations might be 50/50 or 70/30. If the optimal allocations for males are 50/50, how would that effect the predictions? Would one be able to detect an effect? And although it isn’t mentioned in the text, sexually transmitted diseases might enter into the balance of selective forces as well.

Facial symmetry is employed as an indicator of facial attractiveness. The authors portray this symmetry/asymmetry as fluctuating asymmetry. But fluctuating asymmetry is a population parameter. Individual asymmetry may collectively represent fluctuating asymmetry, or it may represent directional asymmetry or antisymmetry or a mixture of all three kinds of asymmetry. The authors mention directional asymmetry when they mention the morphometric analysis and Procrustes distance, but that is the last time it appears. Was there detectable directional asymmetry?

The other two forms of asymmetry (directional asymmetry and antisymmetry) may be better indicators of male quality than fluctuating asymmetry. If low fitness males exhibit fluctuating asymmetry, the modal low fitness males will still be perfectly symmetrical. But if low fitness is associated with a transition from fluctuating asymmetry to directional asymmetry (or antisymmetry), then the predictive value of individual asymmetry is much better. There is a literature for this and evidence for transitions among the three forms of asymmetry in populations.

Throughout the text, the authors mention that fluctuating facial asymmetry is associated with attractiveness. This is a misleading way to portray this association, because it is symmetry, not asymmetry, that is considered attractive. Moreover, it is “individual” symmetry (or asymmetry) that potential mates respond to. Fluctuating asymmetry is assumed to underlie the variation in individual asymmetries. This is an assumption that must be tested, but it rarely is. At the very least it needs to be mentioned.

In addition, the evidence that fluctuating asymmetry is an indicator of genetic quality has been long discussed, and without resolution. It would pay to study (and cite) some of the papers by population geneticists. They generally aren’t sympathetic to the idea of “genetic” quality. How would you measure it? In what sense might a male have “quality” genes? This brings us to Darwinian and inclusive fitness. There are very few studies that have actually looked at fluctuating asymmetry and all possible fitness components (ability to find a mate, fertility, etc.).

For the measurement of fluctuating asymmetry, what is the measurement error associated with the approach? All studies of FA need to assess measurement error, because it can inflate estimates of FA. To do this, the authors need to take more than one photo of each person and then use FACE ++ and MorphoJ to estimate overall asymmetry and follow that with a variance component ANOVA. The methodology in this section is extremely unclear. I presume that a single, unitless number is produced (Procrustes distance), with larger numbers indicating more asymmetry.

For the statistical analysis, the relationship between the Strategic Pluralism Hypothesis and its predictions is unclear. You need to make clear, for example, that the SPH predicts a relationship between socio-sexual orientation and perception of attractiveness and facial symmetry. Moreover, there are no alternative hypotheses mentioned in the paper. If the results didn’t support SPH, what would they support?

The English is grammatically good, though somewhat wordy. In several places, I found the meaning of a phrase or sentence to be obscure. This is usually attributable to wordiness and passive voice.

In the figures, it would be more effective to spell out SOI, FFA, and SPA. I was finding myself having to look these up every time I came across them.

I have also included many comments on the manuscript itself.

Reviewer #2: The authors have conducted a novel test of a well-established theory that adds to knowledge in this area, contains different measures (e.g., biological, psychometric) and is conducted on a decent sized sample that may be deemed ‘non-WEIRD’. The manuscript is of interest to scholars in different fields. The authors find relationships between self-perceived attractiveness, self-perceived fighting ability and openness toward short-term sexual relationships, albeit one that may differ according to the model used (e.g. attractiveness appears to be more stable predictor).

Based on the information reported, I deem there no ‘fatal flaws’ in the work, but would suggest the work requires ‘major revision’ in light of the points below.

General method

I was a little bit concerned that this study appears to be part of an experiment involving economic games - this clearly functions as a competitive task but does not seem to be anywhere in the manuscript (unless I’ve missed something). Personally it seems like something that should be part of the current paper (regardless of whether it constitutes a ‘failed manipulation’) as it could feasibly alter testosterone levels (particularly with the size of the potential reward offered)? A good justification would be required for why this is treated as a ‘separate study’ and/or explanation of how it was part of the general study session. On a second point, you control for relationship status in your analyses but have a sample of men who have responded to items such as ‘….other than your current partner’ when around half the sample are single. I’m not sure if this represents an issue to consider/explain (see next point).

Statistical models and their relation to hypotheses

The way you setup your argument, I’m expecting to see a single model to test the relative/unique contribution of ‘mate quality’ and ‘resource holding potential’ on short term sexual strategies. You sort-of do this (and take into account the different sample sizes when including testosterone in the model). But at no point do you include symmetry in the same model as the predictors related to competitive ability, and I’m not really sure why. The use of the term ‘competitive’ (when describing perceived fighting ability as a proxy for competitiveness, like a personality trait) is a bit odd as people who consider themselves fighters might not be ‘competitive’ per se (i.e., they might not have to try as hard because they are good fighters). Finally on this issue, given the SOI questionnaire (see above) and effects of your control variable (relationship status), I did wonder why the predicted effects were not explored to see if they interacted with partnership status (i.e., stronger among single men, who are more likely to be competing for mates).

Stylistically, I found the analyses hard to follow, which was a problem as there are multiple models, and some effects altered beyond conventional significance in different models/with different control variables. I would suggest reporting the analytical strategy as each model is reported in the results. Then, depending on your response to my earlier point, I would suggest omitting tables and reporting all stats within-text, moving the stats from Table 1 to the appropriate point in the methods (or at the very least, provide a single table with sub-headings to show how the model is ‘built’).

Discussion

Based on above, I would revisit this section for clarity, so that the conclusions match the models, particularly your point on line 307 and line 330 (i.e., you don’t measure reproductive success – only one subscale of the SOI is applicable here and even this could be deemed ‘reproductive potential’ rather than ‘reproductive success’). ‘Three partially supported hypotheses’ is a bit vague/subjective in light of my reading of your results thus far. Line 348 is an important oversight (i.e., why did you examine testosterone if you did not expect it to be related to ‘reproductive strategies’). You then say on line 363 that there is an effect of testosterone! I would also relate the concluding paragraph more directly to the study data.

Minor points.

• Second sentence of abstract (better wording?)

• Line 46 – ‘highly valued’ implies a strong relationship between asymmetry and attractiveness (it exists across studies, but some papers find the relationship tends to be small in effect size).

• Around line 96 – I would suggest brief discussion of some null findings for balance, e.g. work by Michal Kandrik. I would suggest briefly discussing Quist et al. 2012 in the Discussion (as she finds that high SOI women prefer male facial symmetry).

• Line 100 – ‘more interested’ rather than ‘highly interested’ (the latter may imply a strong effect).

• Page 5, first paragraph – I was struggling to follow the logic of the argument here – why many sexual partners would reduce testosterone (and any accompanying citation), why you are arguing that these men are only competing for short-term partners, and why citation # 26 is then followed by discussion of self-perceived fighting ability. Please could you unpack/clarify?

• Just as a general thought - it’s a shame self-rated attractiveness wasn’t anchored to an ‘average’ in the same way that the fighting scale was.

• In the tables, I haven’t heard of the phrase ‘typical error’ – do you just mean ‘standard error’?

• First paragraph of discussion – a bit ‘wordy’ (please clarify/unpack).

General proofreading

Lines 33 (optionally); 36 (fifth region – plus specify roughly where this is – also applies elsewhere with ‘5th’); 66 (electing); 62 (please refer to males rather than men if Trivers is the citation); use of terms ‘relational’ and ‘residue’ (versus ‘relationship status’ and ‘residuals’); 291 (‘tested to determine’); 326 (senses).

Thanks for the opportunity to review this interesting research.

6. PLOS authors have the option to publish the peer review history of their article (what does this mean?). If published, this will include your full peer review and any attached files.

Reviewer #1: No

Reviewer #2: No

---

## [Author Response · Author response to Decision Letter 0]

13 May 2020

RESPONSE TO EDITOR AND REVIEWERS COMMENTS

Dear Dr Polo,

Thank you for submitting your work to PLOS ONE, and apologies for the delay in returning the manuscript to you. I have now secured the opinion of two reviewers in the field. As you will see from their comments, they generally find much to like in the manuscript, commenting on the large sample size, the number of theoretically driven measures, and the testing of an influential theory in the area. They also highlight different concerns about the manuscript, which I agree with. The concerns of Reviewer 1 centred around the presentation of the hypotheses of the SPH, which was presented as an essentially dichotomous outcome for a given male, when in reality it is likely to be more continuous. The bearing on the results and interpretation should be considered. Reviewer 1 also raised concerns over the measurement, definition, and accuracy of the use of facial asymmetry as a proxy for attractiveness perception. This is something I also agree with, and have researched myself (e.g. Jones and Jaeger, 2019; Symmetry). I do not think that these are insurmountable issues and careful discussion and interpretation of the measures is required.

R: We totally agree about the continuous nature of human reproductive strategies. We change some parts in the introduction be clearer about this In addition, we also have addressed the issues about fluctuating asymmetry as a reliable signal of quality and attractiveness. More details about the changes performed can be found in the answers given to reviewer 1.

Reviewer 2 pointed out some more statistical issues with the manuscript, as well as the use of economic games which was puzzlingly mentioned once, and then is not touched upon again. I agree with Reviewer 2 in that much more detail is needed about the background of the work here. In my opinion, and after reading the manuscript, perhaps the most problematic part of the manuscript were the analyses and the modelling procedure. Reviewer 2 also pointed this out and I am in full agreement with them on this point; that there are too many models here testing too many hypotheses.

R: We have provided more details about the background and answered the reviewer 2 concerns about the procedure.

The main issue is that while you have sufficient data and variables to test a model of short-term SOI approaches, splitting variables across into different models obviously ignores the fact that in reality, these variables were measured from, and exist within, the sample of males you obtained. The data-generating process (DGP) is not being accurately tested here. In my view, you have the capacity to run build two models - one which includes all predictors and your full sample, which includes any theoretically important interactions that you wish to test, and another which is includes testosterone as a measure on the reduced sample for which those measures are available for. This way, you can be more certain in your conclusions about what predictors affect what in the presence of other variables, rather than splitting things up. Alternatively, you may wish to first account for the variation explained by age and partner status, and fit your models in a stepwise fashion. Either way, I do not think the current analysis is convincing in its conclusions as it fails to approximate the DGP. I also wondered about the transformations used here. Squaring a variable if it is somewhat non-normal will not make it more normal, but will lead to a kind of gamma-distributed variable (most values bunched up to one side of the distribution), but in any case the mathematics of ordinary least squares does not mind non-normally distributed variables for the predictors or the response; only that the residuals are normally distributed. This is mentioned implicitly but I couldn't tell if the residuals were normal or something was wrong with the predictors.

I was unsure about the use of the mediation analysis here. There are significant reservations about the use of mediation analysis in cross sectional designs - for mediation to be conclusive multiple time points are required, or at least with a cross sectional design heavily theoretical arguments and multiple measures used as mediators should be used to build confidence in the results. I think a far safer conclusion could be made by specifying an interaction between FA and SPA - for example, do men with lower levels of FA perceive themselves to be particularly attractive, and thus have a specific SOI score? To test this would require a simple interaction and centring of your variables. 

R: Thanks for your comments. We have followed your advice and have changed our analyses to avoid mediation. Our original proposal was that more symmetric individuals were expected to show higher short-term sociosexual orientation partially because they perceived themselves as highly attractive. It is reasonable to think that it exists a causal relationship between FA and self-perception of physical attractiveness, although it is true that the later can be affected by multiple factors in addition to FA. A similar reasoning was made for testosterone and self-perception of fighting abilities. In addition, previous studies (e.g. Lukaszewsky et al., 2014), proposed a mediation effects between variables of attractiveness and fighting ability on short-term sociosexual orientation. However, as you mention, mediation analysis with cross-sectional data produces estimators that are biased and this is problematic, especially with full mediation effects. Consequently, we decided to change our proposal and consider now that psychological variables are expected to moderate the effect of FA and testosterone. This decision has produced major changes in the manuscript, but we think that it has gained in methodological rigor.

 1. Lukaszewski AW, Larson CM, Gildersleeve KA, Roney JR, Haselton MG. Condition-dependent calibration of men’s uncommitted mating orientation: evidence from multiple samples. Evol Hum Behav. 2014;35(4):319-26. doi: 10.1016/j.evolhumbehav.2014.03.002

The manuscript was well written and has a high quality data set (which I would encourage the authors share via the OSF, if they can), and I look forward to receiving a resubmission that addresses these concerns.

5. Review Comments to the Author

Reviewer #1: This study gathers together evidence in support of the Strategic Pluralism Hypothesis, the idea that males can allocate time and resources to either mate selection or the raising of children. The study is well done and believable, but the paper can be improved especially in the presentation of the Strategic Pluralism Hypothesis and its predictions.

The Strategic Pluralism Hypothesis is presented first, followed by a set of the sub-hypotheses. These sub-hypotheses are actually predictions of the Strategic Pluralism Hypothesis, but is unclear how they are related. They also need to be portrayed as predictions. For example, if SPH is true, then we expect a, b, and c to be true. Finally, what are the alternatives to the SPH? Science functions best when there are alternative explanations for phenomena.

R: We have clarified and justified the predictions proposed this study based on expectations from the SPH. In addition, based on a suggestion from the editor, we have modified the analytical approach since mediation analysis applied to cross-sectional data may be problematic. The predictions now are based on moderation effects rather than mediation effects. We agree that an alternative hypothesis would enrich the paper, but we did not include them in the paper for several reasons. First, from an evolutionary perspective, the Strategic Pluralism Hypothesis and the Sexual Strategies Theory are the two main theoretical frameworks that aim to explain the variability in human reproductive strategies. However, both proposals are not incompatible rather they focus in different aspects of human reproductive strategies. We selected the SPH because is focused more in within sex variation and flexibility in the behavioral manifestations and we were interested in these aspects in this paper. Second, the other theoretical framework that aim to explain the variability in human reproductive strategies is the Social Role Theory. Unfortunately, our research design was not developed to test the influence of cultural of social norms in the expression of these behaviors as to do so we had needed a sample with a significant variation in cultural background and in the gender roles. Moreover, our study is aimed in within sex variability, and Social Role Theory mainly focuses in between sex differences.

The Strategic Pluralism Hypothesis is posed as an either/or affair, when it is really an entire spectrum. The two ends of the spectrum are 100% of the time and energy spent trying to mate with every available partner versus 100% of the time spent raising children with one partner. But the time and energy allocations might be 50/50 or 70/30. If the optimal allocations for males are 50/50, how would that effect the predictions? Would one be able to detect an effect? And although it isn’t mentioned in the text, sexually transmitted diseases might enter into the balance of selective forces as well.

R: We totally agree that reproductive tactics are a spectrum. Indeed, we decided to employ the SOI questionnaire developed by Jackson & Kirkpatrick because, in our knowledge, it is the only one that considers the attitudes towards unrestricted (short-term) and restricted (long-term) sociosexuality as two independent factors. In this paper, we focused on short-term reproductive tactics and postulated that the traits considered are affecting the predisposition of individuals to invest in short-term partners. Accordingly, we expected a higher unrestricted sociosexuality for those individuals with higher expression of attractive and competitive traits, regardless of their restricted sociosexuality. We have rewritten some parts in the introduction to make clear this point and to add ecological and social features as additional selective forces that play a role in the mentioned trade-off.

Facial symmetry is employed as an indicator of facial attractiveness. The authors portray this symmetry/asymmetry as fluctuating asymmetry. But fluctuating asymmetry is a population parameter. Individual asymmetry may collectively represent fluctuating asymmetry, or it may represent directional asymmetry or antisymmetry or a mixture of all three kinds of asymmetry. The authors mention directional asymmetry when they mention the morphometric analysis and Procrustes distance, but that is the last time it appears. Was there detectable directional asymmetry?

R: We employed Proclustes ANOVA analysis to decomposed variance in shape due to directional and fluctuating asymmetry. Therefore, directional asymmetry was detectable and took into account when calculating individual values of fluctuating asymmetry through this methodology. We have added more information about how we calculated individual values of fluctuating asymmetry in methods.

The other two forms of asymmetry (directional asymmetry and antisymmetry) may be better indicators of male quality than fluctuating asymmetry. If low fitness males exhibit fluctuating asymmetry, the modal low fitness males will still be perfectly symmetrical. But if low fitness is associated with a transition from fluctuating asymmetry to directional asymmetry (or antisymmetry), then the predictive value of individual asymmetry is much better. There is a literature for this and evidence for transitions among the three forms of asymmetry in populations.

R: We agree with referee that there are three kinds of population asymmetries: fluctuating asymmetry, directional asymmetry, and antisymmetry. However, fluctuating asymmetry is the population-level measure of developmental instability, that accounts with the more robust theoretical support as a measure quality. In this sense, all individuals in population exhibit a determined level of fluctuating asymmetry, being the more symmetrical individuals, which possess the values near to zero. Therefore, the higher fitness males possess the lower values in fluctuating asymmetry. Directional asymmetry and antisymmetry usually have been considered as the result of strong genetic effect (e.g., 1). A better example for this is handedness in human being and their effect over biomechanical development of arm muscularity. As we stated before, we have followed the typical procedure of split fluctuating asymmetry form directional asymmetry, we knew that there is new proposal to integrate different kind of symmetry (2), however, we prefer to use the more validated methodology. 

On the other hand, we agree with referee that there is evidence from transitions between fluctuating asymmetry to directional asymmetry in human being, for example for the orbital opening (3), but these analyses are centered in the study of symmetry from the effect of environmental or biological variables. We have not included these predictions in our theoretical model, because we are studying the effect over a determined behavior, and not the causes of fluctuating asymmetry. Therefore, we have not estimated transitions in symmetry, and have used the typical estimation. 

Finally, despite that we consider fluctuating asymmetry as a measure of developmental instability, we mention in the new version of the manuscript that there is controversy about the importance of fluctuating asymmetry as a marker of this instability.

1. Özener, B. (2010). Fluctuating and directional asymmetry in young human males: effect of heavy working condition and socioeconomic status. American journal of physical anthropology, 143(1), 112-120.

2. Ekrami, O., Claes, P., White, J. D., Weinberg, S. M., Marazita, M. L., Walsh, S., ... & Dongen, S. V. (2020). A multivariate approach to determine the dimensionality of human facial asymmetry. Symmetry, 12(3), 348.

3. Tomaszewska, A., Kwiatkowska, B., & Jankauskas, R. (2015). Is the area of the orbital opening in humans related to climate?. American Journal of Human Biology, 27(6), 845-850.

Throughout the text, the authors mention that fluctuating facial asymmetry is associated with attractiveness. This is a misleading way to portray this association, because it is symmetry, not asymmetry, that is considered attractive. Moreover, it is “individual” symmetry (or asymmetry) that potential mates respond to. Fluctuating asymmetry is assumed to underlie the variation in individual asymmetries. This is an assumption that must be tested, but it rarely is. At the very least it needs to be mentioned.

R: Thanks for the comment. In the reviewed manuscript, we explicitly mention that we assume that fluctuating asymmetry is underlying individual differences in asymmetry. In addition, we clarify that it is facial symmetry rather than asymmetry the feature that is considered attractive.

In addition, the evidence that fluctuating asymmetry is an indicator of genetic quality has been long discussed, and without resolution. It would pay to study (and cite) some of the papers by population geneticists. They generally aren’t sympathetic to the idea of “genetic” quality. How would you measure it? In what sense might a male have “quality” genes? This brings us to Darwinian and inclusive fitness. There are very few studies that have actually looked at fluctuating asymmetry and all possible fitness components (ability to find a mate, fertility, etc.).

R: In the new version of the manuscript, we acknowledge the controversial issue about the consideration of fluctuating asymmetry as an indicator of genetic quality. In addition, we have replaced the term genetic quality with a more general term of individual quality.

For the measurement of fluctuating asymmetry, what is the measurement error associated with the approach? All studies of FA need to assess measurement error, because it can inflate estimates of FA. To do this, the authors need to take more than one photo of each person and then use FACE ++ and MorphoJ to estimate overall asymmetry and follow that with a variance component ANOVA. The methodology in this section is extremely unclear. I presume that a single, unitless number is produced (Procrustes distance), with larger numbers indicating more asymmetry.

R: Following to our ethical protocol, we selected one picture of each individual and calculation of fluctuating asymmetry were performed on them. In former studies, two researchers placed the landmarks in each individual (39 landmarks) and we calculated error from this source. This time we decided to employ FACE++ to increase the number of landmarks to 106, but as the points were placed always in the same place for each picture, we lack measurement error.

The methodology has been clarified. Fluctuating asymmetry was calculated as the deviation of each individual’s asymmetry from the overall average asymmetry. To do this first, we calculated Proclustes distances that reflect both directional and fluctuating asymmetry. Then, we employed a Proclustes ANOVA to decompose variance in these two types of asymmetries. Accordingly, we obtained a single measure of fluctuating asymmetry for each individual with larger numbers indicating more fluctuating asymmetry. 

For the statistical analysis, the relationship between the Strategic Pluralism Hypothesis and its predictions is unclear. You need to make clear, for example, that the SPH predicts a relationship between socio-sexual orientation and perception of attractiveness and facial symmetry. Moreover, there are no alternative hypotheses mentioned in the paper. If the results didn’t support SPH, what would they support?

R: We have clarified and justified the predictions proposed this study based on expectations from the SPH. We agree that alternative hypothesis would enrich the paper, but we did not include them in the paper for several reasons. First, from an evolutionary perspective, the Strategic Pluralism Hypothesis and the Sexual Strategies Theory are the two main theoretical frameworks that aim to explain the variability in human reproductive strategies. However, both proposals are not incompatible; rather, the focus is on different aspects of human reproductive strategies. We selected the SPH because it is focused more in within sex variation and flexibility in the behavioral manifestations, and we were just interested in this aspect in this paper. Second, the other theoretical framework that aims to explain the variability in human reproductive strategies is the Social Role Theory. Unfortunately, our research design was not developed to test the influence of cultural of social norms in the expression of these behaviors as to do so we had needed a sample with a significant variation in cultural background and in the gender roles. Moreover, our study is aimed in within sex variability, and Social Role Theory mainly focuses in between sex differences. 

The English is grammatically good, though somewhat wordy. In several places, I found the meaning of a phrase or sentence to be obscure. This is usually attributable to wordiness and passive voice.

R: Thanks for the comment. We have tried to avoid passive voice and wordiness in this new version.

In the figures, it would be more effective to spell out SOI, FFA, and SPA. I was finding myself having to look these up every time I came across them.

R: In this new version, we avoided the use of acronyms.

I have also included many comments on the manuscript itself.

R: Thanks, we followed almost all your comments and suggestions. The exception is L77 in which we mention that men with lower levels of fluctuating asymmetry tend to be more direct in approaching the opposite sex. We agree with your comment, but we do not mention the proximal causes of this behavior so we decided to keep the sentence unchanged. 

Reviewer #2: The authors have conducted a novel test of a well-established theory that adds to knowledge in this area, contains different measures (e.g., biological, psychometric) and is conducted on a decent sized sample that may be deemed ‘non-WEIRD’. The manuscript is of interest to scholars in different fields. The authors find relationships between self-perceived attractiveness, self-perceived fighting ability and openness toward short-term sexual relationships, albeit one that may differ according to the model used (e.g. attractiveness appears to be more stable predictor).

Based on the information reported, I deem there no ‘fatal flaws’ in the work, but would suggest the work requires ‘major revision’ in light of the points below.

General method

I was a little bit concerned that this study appears to be part of an experiment involving economic games - this clearly functions as a competitive task but does not seem to be anywhere in the manuscript (unless I’ve missed something). Personally it seems like something that should be part of the current paper (regardless of whether it constitutes a ‘failed manipulation’) as it could feasibly alter testosterone levels (particularly with the size of the potential reward offered)? A good justification would be required for why this is treated as a ‘separate study’ and/or explanation of how it was part of the general study session. On a second point, you control for relationship status in your analyses but have a sample of men who have responded to items such as ‘….other than your current partner’ when around half the sample are single. I’m not sure if this represents an issue to consider/explain (see next point).

R: The general procedure involved economic games that were aimed to measure cooperative and aggressive tendencies as a part of a wider project. These games were played after the participants answered the questionnaires. Consequently, answers in the questionnaires were unaffected by games. Basal levels of testosterone were measured before participants played the games. But, as you mention, we unknow whether the knowledge of the potential reward, that was mentioned in the ads, affected testosterone levels. We have added more information about the games in the section of method and discuss the potential effect of the reward in levels of testosterone as a limitation of our study.

Regarding the second point, that question is the only one that considers paired individuals in its original formulation. We employed the same question but during the explanation we indicated that individuals that were not paired could answer that question considering how often they fantasize about having sex with women or men (according to their sexual orientations).

Statistical models and their relation to hypotheses

The way you setup your argument, I’m expecting to see a single model to test the relative/unique contribution of ‘mate quality’ and ‘resource holding potential’ on short term sexual strategies. You sort-of do this (and take into account the different sample sizes when including testosterone in the model). But at no point do you include symmetry in the same model as the predictors related to competitive ability, and I’m not really sure why. The use of the term ‘competitive’ (when describing perceived fighting ability as a proxy for competitiveness, like a personality trait) is a bit odd as people who consider themselves fighters might not be ‘competitive’ per se (i.e., they might not have to try as hard because they are good fighters). Finally on this issue, given the SOI questionnaire (see above) and effects of your control variable (relationship status), I did wonder why the predicted effects were not explored to see if they interacted with partnership status (i.e., stronger among single men, who are more likely to be competing for mates).

R: We did not include fluctuating asymmetry in the model including competitive ability because we found that its effect was fully mediated by self-perception of physical attractiveness. In any case, following a suggestion from the editor, we have modified the analytical approach since mediation analysis applied to cross-sectional data may be problematic, especially with regards to full mediation. Then, we investigated moderation effects rather than mediation effects in this new version. In addition, we have reduced the models to two, one considering a reduced data set and including measures of basal testosterone, and another with the full data set and excluding measures of testosterone. Each model was fitted in three successive steps in order to investigate main effects and the predicted interactions.

We agree with your comment about that good fighters may be not show a competitive personality, but our point is that being a good fighter reduces costs involved in intrasexual competition because may be associated to greater chances to win a potential fight or to signals that intimidate rivals. We have tried to clarify this in the text.

Finally, we initially expected that traits related to attractiveness and competitive abilities would be positively associated to short-term sociosexual orientation in single and paired individuals, although global levels of short-term sociosexual orientation could be affected by relationship status. The reason is that individuals in a relationship may be actually in an uncommitted relationship and then, competing and looking for mates. Nevertheless, we explored the interactions between partnership status and the predicted effects and were not significant.

Stylistically, I found the analyses hard to follow, which was a problem as there are multiple models, and some effects altered beyond conventional significance in different models/with different control variables. I would suggest reporting the analytical strategy as each model is reported in the results. Then, depending on your response to my earlier point, I would suggest omitting tables and reporting all stats within-text, moving the stats from Table 1 to the appropriate point in the methods (or at the very least, provide a single table with sub-headings to show how the model is ‘built’).

R: We have reduced the models and we hope that analyses are easier to follow now.

Discussion

Based on above, I would revisit this section for clarity, so that the conclusions match the models, particularly your point on line 307 and line 330 (i.e., you don’t measure reproductive success – only one subscale of the SOI is applicable here and even this could be deemed ‘reproductive potential’ rather than ‘reproductive success’). ‘Three partially supported hypotheses’ is a bit vague/subjective in light of my reading of your results thus far. Line 348 is an important oversight (i.e., why did you examine testosterone if you did not expect it to be related to ‘reproductive strategies’). You then say on line 363 that there is an effect of testosterone! I would also relate the concluding paragraph more directly to the study data.

R: Due to the analytical changes, discussion has suffered a large amount of modifications. We agree, the sentence in line 348 is confusing, we meant to say that we did not find any effect of testosterone conversely of we had expected. We deleted the last part of the sentence to avoid confusions.

Minor points.

• Second sentence of abstract (better wording?)

R: We have tried to improve clarity.

• Line 46 – ‘highly valued’ implies a strong relationship between asymmetry and attractiveness (it exists across studies, but some papers find the relationship tends to be small in effect size).

R: This sentence has been changed.

• Around line 96 – I would suggest brief discussion of some null findings for balance, e.g. work by Michal Kandrik. I would suggest briefly discussing Quist et al. 2012 in the Discussion (as she finds that high SOI women prefer male facial symmetry).

R: Thanks for the comment. We have included null findings and stressed that the effect depends on the context.

• Line 100 – ‘more interested’ rather than ‘highly interested’ (the latter may imply a strong effect).

R:Thanks, we replaced the word.

• Page 5, first paragraph – I was struggling to follow the logic of the argument here – why many sexual partners would reduce testosterone (and any accompanying citation), why you are arguing that these men are only competing for short-term partners, and why citation # 26 is then followed by discussion of self-perceived fighting ability. Please could you unpack/clarify?

R: We have modified this paragraph in order to be clearer in our argument.

• Just as a general thought - it’s a shame self-rated attractiveness wasn’t anchored to an ‘average’ in the same way that the fighting scale was.

R: In this new version, we have centered all the predictor variables to their means.

• In the tables, I haven’t heard of the phrase ‘typical error’ – do you just mean ‘standard error’?

R: Sorry, it was a mistake. We meant to indicate standard error. 

• First paragraph of discussion – a bit ‘wordy’ (please clarify/unpack).

R: We tried to make clear this paragraph.

General proofreading

Lines 33 (optionally); 36 (fifth region – plus specify roughly where this is – also applies elsewhere with ‘5th’); 66 (electing); 62 (please refer to males rather than men if Trivers is the citation); use of terms ‘relational’ and ‘residue’ (versus ‘relationship status’ and ‘residuals’); 291 (‘tested to determine’); 326 (senses).

R: Thank you for the proofreading

Thanks for the opportunity to review this interesting research.

---

## [Decision Letter · Decision Letter 1]

5 Jun 2020

PONE-D-19-27652R1

Testing strategic pluralism: The roles of attractiveness and competitive abilities to understand conditionality in men’s short-term reproductive strategies

PLOS ONE

Dear Dr. Polo,

Thank you for submitting your manuscript to PLOS ONE. After careful consideration, we feel that it has merit but does not fully meet PLOS ONE’s publication criteria as it currently stands. Therefore, we invite you to submit a revised version of the manuscript that addresses the points raised during the review process.

Dear Dr Polo,

Thank you for resubmitting the manuscript, and addressing the comments of the reviewers. Both reviewers have now responded, and there are a small set of outstanding changes to be made regarding some discrepancies with reported statistics and results discussed in previous reviews. These are only minor changes and I feel confident you can address them easily.

Very best wishes, and I hope you are all safe.

Alex

We look forward to receiving your revised manuscript.

Kind regards,

Alex Jones

Academic Editor

PLOS ONE

Reviewers' comments:

Reviewer's Responses to Questions

**Comments to the Author**

1. If the authors have adequately addressed your comments raised in a previous round of review and you feel that this manuscript is now acceptable for publication, you may indicate that here to bypass the “Comments to the Author” section, enter your conflict of interest statement in the “Confidential to Editor” section, and submit your "Accept" recommendation.

Reviewer #1: (No Response)

Reviewer #2: (No Response)

2. Is the manuscript technically sound, and do the data support the conclusions?

Reviewer #1: Yes

Reviewer #2: Yes

3. Has the statistical analysis been performed appropriately and rigorously? 

Reviewer #1: Yes

Reviewer #2: Yes

4. Have the authors made all data underlying the findings in their manuscript fully available?

Reviewer #1: Yes

Reviewer #2: Yes

5. Is the manuscript presented in an intelligible fashion and written in standard English?

Reviewer #1: Yes

Reviewer #2: Yes

6. Review Comments to the Author

Reviewer #1: The manuscript is much improved over the original.

I have to respectfully disagree with this statement: “However, fluctuating asymmetry is the population-level measure of developmental instability, that accounts with the more robust theoretical support as a measure quality. In this sense, all individuals in population exhibit a determined level of fluctuating asymmetry, being the more symmetrical individuals, which possess the values near to zero. Therefore, the higher fitness males possess the lower values in fluctuating asymmetry. Directional asymmetry and antisymmetry usually have been considered as the result of strong genetic effect (e.g., 1).”

Imagine two populations: high-fitness individuals (blue) and low-fitness individuals (orange). The two curves differ in their variances. But in both cases, the modal individuals are still perfectly symmetrical. There are just fewer perfectly symmetrical individuals among the low-fitness population. Now imagine a transition from FA (blue curve) to antisymmetry (bimodal distribution) in the low-fitness individuals. If the low-quality individuals have an antisymmetric distribution of individual asymmetries, then most individuals are asymmetrical. There is abundant evidence for such transitions, even in the very first paper on fluctuating asymmetry by Kenneth Mather. When Mather selected for increased asymmetry in Drosophila, the population transitioned from FA to antisymmetry. And the heritability of antisymmetry and directional asymmetry is only slightly greater than that of fluctuating asymmetry, so the statement that such asymmetries are “the result of strong genetic effect” doesn’t hold up to scrutiny. See the very detailed papers by Larry Leamy. In some of my own (unpublished) research I’ve been able to document a transition from FA to antisymmetry by knocking down (RNAi) the activity of a single gene. These were in inbred lines with little genetic variation.

[See the figure in the attached file.]

Regardless, the authors need to document that the statistical distributions they are dealing with are symmetrical, unimodal, and with a mean of zero. Does a normal distribution fit the data? One just cannot wish away these other forms of asymmetry, and they may be important.

Reviewer #2: I am happy that the authors have addressed my concerns, except for the following relatively minor points:

• For completeness, please could you include brief critical details on the photography procedure, term the payment as ‘reimbursement’ rather than ‘incentive’, and provide (in-text) the age characteristics of the smaller sample who had testosterone measured.

• The manuscript reads well. One more proofread would be beneficial, e.g., for typos (e.g., ‘physical’ misspelt) wording that is ‘hyperbolic’ (e.g. describing something as having a ‘great’ influence when the effect may be small/moderate), and three overly-complex passages:

“according to the expression of features dependent on the individual's condition, such as a selective response to the reproductive trade-off”

“but there are variables linked to attractiveness (such as self-perception) that have a greater effect in terms of intersexual selection for the case of unrestricted strategies”

I also couldn’t understand the final concluding paragraph (except for its first sentence) – which also refers to mediation, which you no longer do.

Hypothesis 1 also seems oddly worded: “fluctuating asymmetry should be negative associated with short-term reproductive strategies, especially in individuals with high levels of self-perceived physical attractiveness…” – by that I mean it would read better if it referred to ‘facial symmetry’ – as I think you’re trying to get across here that this POSITIVE relationship (with symmetry) would be stronger in individuals who think of themselves as attractive (as they are better able to offset any costs of engaging in short-term mating competition)?

• You explained in the response that single individuals could answer one of the items differently on the SOI, which seems OK. I don’t think you report this in the manuscript, though. Related to this (and my earlier point on partnership status), please carefully do a final proofing check on analyses in light of the major changes made to the manuscript. For example, you’ve said in the response document that there were no effects of relationship status, but there does appear to be an effect in Table 3.

• I leave this as the Editor’s decision, but personally I thought the in-text results could be even more concise – as you seem to be reporting everything in tables, perhaps the text could simply refer to what was/was not significant, cross-reference to the tables, and save page space by not reporting sets of statistical values twice. Please could you also format (italicize) all statistical values.

• Apologies if I’ve missed this, but do you refer anywhere to the scale end points for facial fluctuating asymmetry (and what high/low scores mean), just to get a general sense when reading it of how variable the sample were (variability seems quite low according to the descriptive statistics).

• It might be useful to mention around limitations, briefly, that any null effects of basal testosterone in the current study don’t necessarily rule out relationships between T changes and sexual/competitive behaviours (e.g. if elicited experimentally via a confederate or other manipulation). Please also very briefly mention, in light of my prior comments, that the later competitive tasks might represent a small limitation introducing noise into the current study (i.e., they know they are attending at some point to engage in a competitive task and are being reimbursed a reasonable amount to do so, albeit these tasks are after the measures taken for the current study).

7. PLOS authors have the option to publish the peer review history of their article (what does this mean?). If published, this will include your full peer review and any attached files.

Reviewer #1: No

Reviewer #2: No

---

## [Author Response · Author response to Decision Letter 1]

20 Jul 2020

RESPONSE TO REVIEWERS

PONE-D-19-27652R1

Testing strategic pluralism: The roles of attractiveness and competitive abilities to understand conditionality in men’s short-term reproductive strategies

6. Review Comments to the Author

Reviewer #1: The manuscript is much improved over the original.

I have to respectfully disagree with this statement: “However, fluctuating asymmetry is the population-level measure of developmental instability, that accounts with the more robust theoretical support as a measure quality. In this sense, all individuals in population exhibit a determined level of fluctuating asymmetry, being the more symmetrical individuals, which possess the values near to zero. Therefore, the higher fitness males possess the lower values in fluctuating asymmetry. Directional asymmetry and antisymmetry usually have been considered as the result of strong genetic effect (e.g., 1).”

Imagine two populations: high-fitness individuals (blue) and low-fitness individuals (orange). The two curves differ in their variances. But in both cases, the modal individuals are still perfectly symmetrical. There are just fewer perfectly symmetrical individuals among the low-fitness population. Now imagine a transition from FA (blue curve) to antisymmetry (bimodal distribution) in the low-fitness individuals. If the low-quality individuals have an antisymmetric distribution of individual asymmetries, then most individuals are asymmetrical. There is abundant evidence for such transitions, even in the very first paper on fluctuating asymmetry by Kenneth Mather. When Mather selected for increased asymmetry in Drosophila, the population transitioned from FA to antisymmetry. And the heritability of antisymmetry and directional asymmetry is only slightly greater than that of fluctuating asymmetry, so the statement that such asymmetries are “the result of strong genetic effect” doesn’t hold up to scrutiny. See the very detailed papers by Larry Leamy. In some of my own (unpublished) research I’ve been able to document a transition from FA to antisymmetry by knocking down (RNAi) the activity of a single gene. These were in inbred lines with little genetic variation.

[See the figure in the attached file.]

Regardless, the authors need to document that the statistical distributions they are dealing with are symmetrical, unimodal, and with a mean of zero. Does a normal distribution fit the data? One just cannot wish away these other forms of asymmetry, and they may be important.

R: Thanks for your comment. We have analyzed the distribution of differences between the original and the reflected landmarks from each individual from the coordinates in the asymmetry component provided by the MorphoJ after the Procrustes fit. For the x’s coordinates, we found that 44 out of 48 distributions were normally distributed and for the y’s coordinates, 45 out of 48 distributions were normally distributed. We mentioned these distributions in the methods section. Most of the distributions were not around 0 (40 out 48 for the x-axis and 35 out 48 for the y-axis). This result is in accordance with the significant effect of “side” in the Procrustes ANOVA analysis denoting the existence of directional asymmetry; however, our measure of fluctuating asymmetry took into account the directional asymmetry as we describe in the method section. Normality tests, descriptives and t-tests are reported in the attached file. We appreciate your critical feedback and now the manuscript reflects more accurately the controversy about the significance of fluctuating asymmetry as a reliable indicator of quality. In addition, it is relevant to establish that we have not found articles were antisymmetry was linked to behaviour in human beings. However, there is evidence that supports the presence of a certain degree of antisymmetry from skulls; therefore, we agree with the decision to include these analyses in the manuscript. 

Reviewer #2: I am happy that the authors have addressed my concerns, except for the following relatively minor points:

• For completeness, please could you include brief critical details on the photography procedure, term the payment as ‘reimbursement’ rather than ‘incentive’, and provide (in-text) the age characteristics of the smaller sample who had testosterone measured.

R: Thanks for your suggestions. We have added information about the focal length, speed, aperture and light conditions in which the pictures were taken. We have changed the word incentive and provide more information about age in the reduced data set.

• The manuscript reads well. One more proofread would be beneficial, e.g., for typos (e.g., ‘physical’ misspelt) wording that is ‘hyperbolic’ (e.g. describing something as having a ‘great’ influence when the effect may be small/moderate), and three overly-complex passages:

R: A general proofread has been made. 

“according to the expression of features dependent on the individual's condition, such as a selective response to the reproductive trade-off”

R: We have deleted the last part of the sentence. Our intention was to signal that facultative calibration of reproductive strategies can be understood as an adaptative response to the trade-off, but it is not critical information and deleting it reduced the complexity of the sentence.

“but there are variables linked to attractiveness (such as self-perception) that have a greater effect in terms of intersexual selection for the case of unrestricted strategies”

R: We have amended the last part of this paragraph since some of the argumentation was related to former results. 

I also couldn’t understand the final concluding paragraph (except for its first sentence) – which also refers to mediation, which you no longer do.

R: I am confused about your comment here. First, because in the first sentence of the last paragraph, we do not mention mediation. We mention mediation in the last sentence of the paragraph (maybe you meant to say last instead of first?), but here our point is to signal that designs that are more suitable to study mediation effects may contribute to understand the role of fluctuating asymmetry in future studies. Anyway, we have tried to be clearer in this paragraph.

Hypothesis 1 also seems oddly worded: “fluctuating asymmetry should be negative associated with short-term reproductive strategies, especially in individuals with high levels of self-perceived physical attractiveness…” – by that I mean it would read better if it referred to ‘facial symmetry’ – as I think you’re trying to get across here that this POSITIVE relationship (with symmetry) would be stronger in individuals who think of themselves as attractive (as they are better able to offset any costs of engaging in short-term mating competition)?

R: We agree that it could be less confusing the use of symmetry instead of fluctuating asymmetry, however, we prefer to keep the term fluctuating asymmetry in the predictions because there are two more sources of asymmetry (directional and antisymmetry) and consequently, fluctuating asymmetry and facial symmetry are not 100% equivalent. In the introduction, we mention facial symmetry when talking about the relationship between symmetry and attractiveness, which includes fluctuating asymmetry. 

• You explained in the response that single individuals could answer one of the items differently on the SOI, which seems OK. I don’t think you report this in the manuscript, though. Related to this (and my earlier point on partnership status), please carefully do a final proofing check on analyses in light of the major changes made to the manuscript. For example, you’ve said in the response document that there were no effects of relationship status, but there does appear to be an effect in Table 3.

R: Regarding the question about sociosexual desire “How often do you fantasize about having sex with someone other than your current dating partner?” I agree with you that seems to be intended to paired individuals (indeed I was imprecise in my first response), but the main aim of this item is to assess sexual attraction that is specifically targeted at potential mates to whom no committed romantic relationship exists. Since many single participants asked us that if they had to answer the question, we included in the general explanation a particular clarification on this point. We did not mention that because we only considered the short-term factor of the sociosexual attitude dimension due to the aim of this study.

Regarding relationship status, sorry if my previous response was not clear, but we meant to indicate that we explore potential interactions between relationship status and predictor variables (fluctuating asymmetry, self-perceived psychical attractiveness, basal levels of testosterone and self-perceived fighting ability) and we did not find any interaction effect. The result in Table 3 is related to the main effect of relationship status. In any case, we have carefully checked our analysis, and no inconsistencies in the results were found.

• I leave this as the Editor’s decision, but personally I thought the in-text results could be even more concise – as you seem to be reporting everything in tables, perhaps the text could simply refer to what was/was not significant, cross-reference to the tables, and save page space by not reporting sets of statistical values twice. Please could you also format (italicize) all statistical values.

R: Thanks, we have amended the format of statistical values. 

• Apologies if I’ve missed this, but do you refer anywhere to the scale end points for facial fluctuating asymmetry (and what high/low scores mean), just to get a general sense when reading it of how variable the sample were (variability seems quite low according to the descriptive statistics).

R: Facial fluctuating asymmetry was measured as the deviation of each individual’s asymmetry from the overall average asymmetry. This deviation represents a distance (unitless) in Euclidean space (Procrustes distance), therefore the minimum value is 0 (no deviation from the overall average asymmetry) and the value increases with the deviation from the overall average asymmetry. We have clarified the scale for facial fluctuating asymmetry in the manuscript.

• It might be useful to mention around limitations, briefly, that any null effects of basal testosterone in the current study don’t necessarily rule out relationships between T changes and sexual/competitive behaviours (e.g. if elicited experimentally via a confederate or other manipulation). Please also very briefly mention, in light of my prior comments, that the later competitive tasks might represent a small limitation introducing noise into the current study (i.e., they know they are attending at some point to engage in a competitive task and are being reimbursed a reasonable amount to do so, albeit these tasks are after the measures taken for the current study).

R: We have included the suggested limitation in our discussion.

We appreciate all your critical comments and suggestions.

---

## [Editor Report · Decision Letter 2]

24 Jul 2020

Testing strategic pluralism: The roles of attractiveness and competitive abilities to understand conditionality in men’s short-term reproductive strategies

PONE-D-19-27652R2

Dear Dr. Polo,

We’re pleased to inform you that your manuscript has been judged scientifically suitable for publication and will be formally accepted for publication once it meets all outstanding technical requirements.

Kind regards,

Alex Jones

Academic Editor

PLOS ONE
---

## [Editor Report · Acceptance letter]

14 Aug 2020

PONE-D-19-27652R2 

Testing strategic pluralism: The roles of attractiveness and competitive abilities to understand conditionality in men’s short-term reproductive strategies 

Dear Dr. Polo:

I'm pleased to inform you that your manuscript has been deemed suitable for publication in PLOS ONE. Congratulations! Your manuscript is now with our production department. 

Kind regards, 

on behalf of

Dr. Alex Jones 

Academic Editor

PLOS ONE